# Retraction of Clinical Trials about the SARS-CoV-2 Infection: An Unaddressed Problem and Its Possible Impact on Coronavirus Disease (COVID)-19 Treatment

**DOI:** 10.3390/ijerph20031835

**Published:** 2023-01-19

**Authors:** Felipe Eduardo Valencise, Camila Vantini Capasso Palamim, Fernando Augusto Lima Marson

**Affiliations:** 1Laboratory of Cell and Molecular Tumor Biology and Bioactive Compounds, São Francisco University, Bragança Paulista 12916-900, SP, Brazil; 2Laboratory of Human and Medical Genetics, São Francisco University, Bragança Paulista 12916-900, SP, Brazil

**Keywords:** clinical trial, mortality, pandemic, SARS-CoV-2, science

## Abstract

We are presenting an overview of the retracted clinical trials about the Coronavirus Disease (COVID)-19 published in PubMed using the descriptors ((COVID-19 OR SARS-CoV-2) AND (Clinical Trial)). We collected the information for i) the first author’s country; ii) the journal name where the study was published; iii) the impact factor of the journal; iv) the main objective of the study; v) methods including population, intervention, study design, and outcomes; and vi) results and conclusions. We collected complete information from the retraction notes published by the journals and the number of publications/retractions related to non-COVID-19 clinical trials published simultaneously. We also included the Altmetric index for the clinical trials and the retraction notes about COVID-19 to compare the accessibility to both studies’ indexes. The retraction of clinical trials occurred in four countries (one in Lebanon, one in India, one in Brazil, and five in Egypt) and six journals (one in Viruses, one in Archives of Virology, one in Expert Review of Anti-infective Therapy, one in Frontiers in Medicine, two in Scientific Reports, and two in The American Journal of Tropical Medicine and Hygiene). Eight drugs were tested (Ivermectin, Vitamin D, Proxalutamide, Hydroxychloroquine, Remdesevir, Favipiravir, and Sofosbuvir + Daclatasvir) in the studies. One of the retractions was suggested by the authors due to an error in the statistical analysis, which compromised their results and conclusions. Also, the methods, mainly the allocation, were not well conducted in the two studies, and the studies were retracted. In addition, the studies performed by Dabbous et al. presented several issues, mainly including several raw datasets that did not prove their findings. Moreover, two studies were retracted due to data overlap and copying. Significant concerns were raised about the integrity of the data and reported results in another article. We identified a higher Altmetric index for the original studies, proving that the retracted studies were accessed more than the retraction notes. Interestingly, the impact of the original articles is much higher than their retraction notes. The different Altmetric indexes show that possibly people who read those retracted articles are not reading their retraction notes and are unaware of the erroneous information they share. COVID-19- related clinical trials were ~two-time times more retracted than the other clinical trials performed during the same time.

## 1. Introduction

Coronavirus Disease (COVID)-19 is the most widespread and severe pandemic in modern times. The world, including the scientific community, turned their efforts to investigate the disease, searching for pathophysiological mechanisms, risk factors, possible treatments, and vaccines. Following the “publish or perish” mantra, the urge for discoveries regarding COVID-19 has taken its toll on evidence-based science, resulting in a significant number of papers published about COVID-19 [1]. Valencise et al. (2022) showed the impact COVID-19 caused on the publication of papers regarding the 25 leading death causes, according to world region, implicating that while everyone was focused on COVID-19-related research, the leading causes of death had their research postponed or canceled [2].

With such a fast income of COVID-19 publications, followed by their retractions, the characteristics of the removed papers can provide important details into the nature of the removals and the review process [3,4]. Articles may be retracted when their findings are not trustworthy. This can occur due to scientific misconduct or error (e.g., data manipulation, fraudulent data, unsupported conclusions, questionable data validity, non-replicability, data errors—even if unintended) or ethical guidelines violations (e.g., duplicate publication, plagiarism, missing credit, no institutional review board, ownership issues, authorship issues) [5,6,7]. In first-quartile journals like The Lancet and The New England Journal of Medicine, retractions of papers from journals like these are in the minority in retraction but raise a more significant concern for evidence-based medicine [3]. Moreover, it is becoming harder to understand the reasons behind the removal of the papers, making the importance of the retraction note significantly higher. In addition, due to the high number of published papers, it is difficult to read all of them, even the retractions notes.

In this matter, several articles were rushed and published under poor evaluation, causing the number of retractions to escalate vigorously [8], including the clinical trials studies about COVID-19, which comprised the utmost importance articles, causing the need for several retraction notes [9,10,11,12,13,14,15,16]. In theory, once a paper is retracted, it should only be cited by other studies in the context of their retraction [7,17]. However, in reality, there are an impressive number of citations on retracted papers without mentioning the retraction of said papers, which raised several concerns in the scientific community [7,17]. Furthermore, not only is rushing papers to publication, sometimes under poor evaluation, dangerous, the retraction notes provided by the journals bring very little information regarding the reasons for retraction and do not get as many visualizations as the original article, which implies that not all people that read the paper got access to the retraction note.

This problem gets aggravated during a pandemic such as COVID-19 once everyone searches for pathogenesis and possible treatments. Rush-publishing papers in this scenario impact the quality of treatment for the disease, making it possible for medications that were not properly evaluated to gain momentum in the great masses, disseminating unverified information [18].

In this context, we aimed to present an overview of the retracted clinical trials during the COVID-19 pandemic published in PubMed and to discuss why this issue was aggravated during the pandemic, calling for attention to this unaddressed problem, its impact on COVID-19 treatment and why it may put science in jeopardy once it stimulates quantity over quality.

## 2. Materials and Methods

In the present brief report, we present an overview of the retracted clinical trials during the COVID-19 pandemic and for COVID-19 interventions published in PubMed using the descriptors ((COVID-19 OR SARS-CoV-2) AND (Clinical Trial)). PubMed database was selected to standardize the study because PubMed database uses an algorithm that searches the title, abstract, and headings of articles in the National Library of Medicine database—specific to medicine and health [19]. Also, we summarize clinical trials published during the COVID-19 pandemic (from the year 2019 to 12 December 2022) not related to COVID-19 published in PubMed using the descriptors ((Clinical Trial) NOT (COVID-19 OR SARS-CoV-2)). We presented the papers as the indexation type: “clinical trial Phase I”, “clinical trial Phase II”, “clinical trial Phase III”, “clinical trial Phase IV”, “clinical trial protocol”, and “randomized clinical trial”. We presented the proportion of each type of paper by the total number of clinical trials published. In addition, we selected the types of papers classified only as “retracted publication” or “retraction of publication” (COVID-19-related) to present a complete description of their findings. We excluded the papers not associated with COVID-19 interventions.

In this context, considering only the retracted papers for COVID-19-related interventions, we collected the information for (i) the first author’s country; (ii) the journal name where the study was published; (iii) the impact factor of the journal; (iv) the main objective of the study; (v) methods including population, intervention, study design, and outcomes; and (vi) results and conclusions. We also collected complete information from the retraction notes published by the journals, the Altmetric index for the clinical trials and the retraction notes to compare the accessibility to both studies’ indexes (https://www.altmetric.com/ accessed on 16 December 2022). In addition, the number of citations achieved from the PubMed database was described.

The Altmetric is an index that collects and collates all of the disparate information about research to provide the Scientific community with a visually engaging and informative view of the online activity surrounding the scholarly content. In brief, as described by the developers, “*Altmetrics are metrics and qualitative data that are complementary to traditional, citation-based metrics. They can include (but are not limited to) peer reviews on Faculty of 1000, citations on Wikipedia and in public policy documents, discussions on research blogs, mainstream media coverage, bookmarks on reference managers like Mendeley, and mentions on social networks such as Twitter. Sourced from the Web, Altmetrics can tell you a lot about how often journal articles and other scholarly outputs like datasets are discussed and used around the world. For that reason, Altmetrics have been incorporated into researchers’ websites, institutional repositories, journal websites, and more.*”.

We compared the proportion between the number of retracted clinical trials by the total number of published clinical trials considering the ratio between COVID-19-related papers and other clinical trials.

## 3. Results

On 12 December 2022, a total of 8445 studies were published in PubMed using the descriptors ((COVID-19 OR SARS-CoV-2) AND (Clinical Trial)). From them, 142 were Phase I studies, 250 were Phase II studies, 156 were Phase III studies, and 17 were Phase IV studies. In addition, 2086 studies were indexed as randomized controlled trials and 549 as clinical trial protocols (Table 1). Also, using the descriptors, nine studies were retracted [9,10,11,12,13,14,15,16,20]; however, one study was not associated with a COVID-19 intervention, and it was excluded, as well as its retraction note [20] (Table 1 and Table 2). During the same period, from January 2019 to December 12, 2022, we observed a total of 202,398 studies published in PubMed using the descriptors ((Clinical Trial) NOT (COVID-19 OR SARS-CoV-2)). From them, 4774 were Phase I studies, 7426 were Phase II studies, 5445 were Phase III studies, and 594 were Phase IV studies (Table 1). In addition, 92,111 studies were indexed as randomized controlled trials and 7942 as clinical trial protocols. Also, 111 clinical trial studies were retracted (Table 1).

Until 12 December 2022, the COVID-19-related articles (clinical trials) were 1.73 times (8/8445 by 111/202,398) more retracted than other clinical trials during the same period (Table 1). Curiously, clinical trial protocols were proportionally more published for COVID-19-related clinical trials than non-COVID-19-related clinical trials; in contrast, all clinical trial phases and the descriptor randomized clinical trial were more evident for non-COVID-19-related clinical trials.

The retraction of clinical trials occurred in four countries (one in Lebanon, one in India, one in Brazil, and five in Egypt) and six Journals (one in Viruses, one in Archives of Virology, one in Expert Review of Anti-infective Therapy, one in Frontiers in Medicine, two in Scientific Reports, and two in The American Journal of Tropical Medicine and Hygiene) (Table 2).

In Table 3, we presented the characterization of the clinical trials about COVID-19 retracted after publication, including their objective, methods (intervention, study design, and outcomes), results, and conclusions. In addition, we presented the retraction notes and the Altmetrics indexes (crude data) for the clinical trials (original study) and their retraction note. In Figure 1, we presented the number of citations of the clinical trials (original study) and their retraction notes. Curiously, all clinical trials were cited, and two received more than 100 citations [13,15]. Besides that, only four retractions notes were cited by eight, one, five, and three studies [9,10,12,13]. Between both studies that received more than 100 citations, only one had citations for the retraction note [13]. Also, the retraction note with the highest number of citations (eight) was associated with one study with 37 citations [9]. 

One of the retractions was suggested by the authors due to an error in the statistical analysis, which compromised their results and conclusions [9]. Also, the methods, mainly the allocation, were not well done in the two studies, and the studies were retracted [10,11]. In addition, the studies performed by Dabbous et al. (2021, 2022) [12,13] presented several issues, mainly including several raw datasets that did not prove their findings. Moreover, two studies were retracted together due to data overlap and/or copying; in this case, the authors have not provided a reasonable explanation for this significant problem, and the authors have not provided adequate data error-checking or validation to ensure that the remaining results presented in the paper accurately represent the sourced data [14,15]. Also, significant concerns have been raised about the integrity of the data and reported results in one article; the authors have been unable to address the concerns raised fully and cannot provide sufficient supporting information [16] (Table 3).

Figure 2 presents the Altmetrics indexes for clinical trials (original studies) and their retraction notes. In our data, there is a high amplitude between the Altmetrics indexes for the clinical trials, which ranged between one [16] to 1763 [10] (Figure 2a); however, the amplitude was low among the retractions notes which ranged between zero [13,16] and 344 [9] (Figure 2b). In addition, we calculated the proportion between the Altmetrics indexes for the clinical trials and their retraction notes (Figure 2c). The clinical trial with the highest Altmetric index had a higher ratio (16.17×) [10]. To note, two clinical studies had a higher Altmetric index for the retraction note [12,14], and maybe one of them can be related to the fact that the retraction note [14] was made for two simultaneous clinical trials [14,15].

Of those retracted papers, one evaluated Ivermectin (PubChem CID: #6321424; C_48_H_74_O_14_) as a potential agent to control the viral load of SARS-CoV-2 among asymptomatic participants in Lebanon [9]. The paper was later retracted because the authors identified an error in files during the statistical analysis, which comprised the study and its findings [9]. Initially, the authors described that Ivermectin could cause fewer symptoms, lower viral load, and reduce hospital admissions in patients infected with SARS-CoV-2 but asymptomatic on inclusion.

A third clinical trial was retracted, and it evaluated the inclusion of Pulse D therapy to reduce the inflammatory markers of COVID-19 [10]. In this study, the authors described that vitamin D levels increased after Pulse D therapy in the vitamin D group and were associated with a reduction of the measured inflammatory markers. In addition, the authors identified a significant difference in the decrease in inflammatory markers between the study groups [10]. The study was later retracted because significant differences in the baseline parameters were identified, indicating that randomization may not have been performed correctly [10]. In this context, a post-publication peer review identified that the allocation method used was not appropriate. In brief, the study was retracted because the methods were unsuitable for affirming that Pulse D therapy was the only factor associated with the different outcomes between both study groups [10].

An American first author conducted a Brazilian study to describe if Proxalutamide (PubChem CID: #60194102; C_24_H_19_F_4_N_5_O_2_S), an androgen receptor antagonist, was effective in treating men with COVID-19 in an outpatient setting [11]. In the study, the authors included 268 men that received Proxalutamide (*N* = 134) or placebo (*N* = 134). In brief, the authors noticed that the 30-day hospitalization rate was 2.2% in men taking Proxalutamide when compared to 26% in placebo [11]. After the publication, several concerns were raised about the methods, and the study was retracted mainly due to errors in the allocation process, which was not random [11].

Two studies were performed in Egypt by Dabbous et al. (2021, 2022) [12,13] to evaluate the benefits of Favipiravir (PubChem CID: #492405; C_5_H_4_FN_3_O_2_) versus Hydroxychloroquine (PubChem CID: #3652; C_18_H_26_ClN_3_O) and oral Oseltamivir (PubChem CID: #65028; C_16_H_28_N_2_O_4_) in managing patients with COVID-19. The first study concluded that Favipiravir was a safe and effective alternative to Hydroxychloroquine in mild or moderate SARS-CoV-2 infected participants [12]. However, after the publication, several concerns were raised by the scientific community, and the editors requested the raw data. Notably, the authors sent several datasets, and none presented the same data and results as those published. In addition, it was demonstrated that the randomization procedure was not entirely performed at random, e.g., the distribution of male and female patients was equal between the groups. However, it is unlikely because sex was not considered during the allocation as a covariate [12]. The second study presented a similar conclusion to the first one. It concluded that Favipiravir is a promising drug for the treatment of patients with COVID-19 that might decrease the duration of the hospital stay and the need for mechanical ventilation [13]. In addition, the study was retracted due to the divergences in the raw data and the differences between the groups at the baseline for several features [13].

Another two retracted clinical trial studies performed in Egypt were included in our review study [14,15]. The first study aimed to assess the efficacy of Remdesivir (PubChem CID: #121304016; C_27_H_35_N_6_O_8_P) in hospitalized Egyptian patients with COVID-19. It concluded that Remdesivir positively influenced the length of hospital stay, but it had no mortality benefit in Egyptian patients with COVID-19 [14]. The second study aimed to evaluate the safety and efficacy of Hydroxychloroquine added to standard care in patients with COVID-19 [15]. The authors found that the overall mortality did not differ between the two groups. Univariate logistic regression analysis showed that Hydroxychloroquine treatment was not significantly associated with decreased mortality [15]. In brief, both studies were retracted in the same retraction note, mainly due to data overlap and/or copying between the manuscripts [14,15].

Finally, one more study from Egypt was included in our review [16]. The study aimed to evaluate the efficacy of generic Sofosbuvir/Daclatasvir (PubChem CID: #45375808; C_22_H_29_FN_3_O_9_P/PubChem CID: #25154714; C_40_H_50_N_8_O_6_) in treating COVID-19 patients with pneumonia [16]. In the study, the authors observed a lower mortality rate in the group that received Sofosbuvir/Daclatasvir. After one month of therapy, no differences were found in intensive care unit admission rates, oxygen therapy, or ventilation support. Additionally, a statistically significant shorter duration of hospital stays (9 vs. 12%) and a faster achievement of polymerase chain reaction negativity at day 14 (84 vs. 47%) were noticed in the treatment group [16]. The article was retracted due to significant concerns raised regarding the integrity of the data and reported results in the article [16].

In brief, the retraction of clinical trials for COVID-19 occurred in four countries (one in Lebanon, one in India, one in Brazil, and five in Egypt) and in six scientific journals (one in Viruses, one in Archives of Virology, one in Expert Review of Anti-infective Therapy, one in Frontiers in Medicine, two in Scientific Reports, and two The American Journal of Tropical Medicine and Hygiene). In total, eight drugs were tested (Ivermectin, Vitamin D, Proxalutamide, Hydroxychloroquine, Remdesevir, Favipiravir, and lastly, Sofosbuvir combined with Daclatasvir) [9,10,11,12,13,14,15,16].

Moreover, we identified a higher Altmetric index for the original studies, proving that the retracted studies were read more than the retraction notes. Interestingly, the impact of the original articles is much higher than their retraction notes. The Altmetrics indexes show that people who read those retracted articles are not reading their retraction notes and are unaware of the erroneous information they share.

We briefly explained the paper retraction (causes and implications) in Figure 3.

## 4. Discussion

It is worrisome that the number of retractions in the era of the COVID-19 pandemic is high [3,21,22], even for clinical trials [9,10,11,12,13,14,15,16]. Yeo-Teh & Tang (2021) described that a PubMed search showed 7440 COVID-19-related articles as of 3rd May 2020. This number escalated to 17,559 articles as of 8th June 2020, leading to exponential growth in publication during a short period [3]. The retractions from clinical trials can compromise the politics of treating patients with COVID-19, mainly in countries with intense scientific disbelief, such as Brazil [18,23].

In the literature, Frampton, Woods, & Scott (2021) identified several problems in how COVID-19 papers were retracted, like lack of clarity on the timing of and reasons for retractions, and continued availability of retracted articles, often from multiple sources, which raises attention on how difficult it is to prove the facts behind the publication of clinical trials and their retraction, which included the implementation of a fast review process and the review by inexperienced reviewers compromising the quality of the review process, the implementation of good study design to give the correct randomization reducing the influence of baseline features into the outcomes, error during the data collection and the statistical analysis, or the misconduct of the authors [22]. However, suppose it is difficult to determine the causes of the publication followed by its retraction. In that case, it is even more challenging to decide whether the published papers were correctly performed or how this type of publication affected health measures. For example, among the cited papers, several were used and cited in systematic and meta-analysis studies, even after retraction. Importantly, Kataoka et al. investigated in a meta-epidemiological study if the systematic reviews and clinical practice guidelines, which included retracted randomized controlled trials, performed a correction letter [24]. Curiously, only 5% of the studies published before the retraction corrected or retracted their results. In addition, several studies cited already retracted randomized controlled trials after publication. Sadly, a significant part of the articles cited those studies in the evidence analysis and did not consider the retraction note, even in the discussion section. We need to better prepare health students and health professionals to read and interpret a scientific article, mainly a publication of a clinical trial. In that sense, the student and professional will be able to perform their conclusion based on the study findings, and they will create a critical view of science.

One question remains: why were the retraction rates on COVID-19 papers higher during the pandemic? After consideration, the authors ended up with three possible reasons for this phenomenon:(1)This was due to pressure to publish, which, despite not being demonstrated empirically for any of these articles, is a valid possibility and further studies are to elucidate this possibility.(2)That retractions are due to insufficient peer review, which implies that skilled reviewers can detect randomization errors, research misconduct, and similar minor errors resulting in work that is not reproducible. It is unclear whether skilled and experienced reviewers would request raw data from the studies or be able to detect some of the issues identified as the cause of retractions upon review.(3)That the rapid influx of papers causes retractions. In this sense, due to the urgency for new information and possible treatment for a new deadly disease, all researchers turned to COVID-19. This converged in a high number of articles being submitted simultaneously.

The main limitations of our study include that the authors only analyzed eight retracted articles, which is a low number when compared to the immense number of published articles during the COVID-19 pandemic. But, despite the low number of retracted articles analyzed, this paper calls for attention to a severe problem. Also, proportionally, there is double the number of papers retracted that are related to COVID-19 interventions than non-COVID-19-related clinical trials during the same period. Moreover, very little information was available regarding the retraction note and its justifications. In addition, it is difficult to assess the quality of clinical trials, including randomized controlled trials, after publication, and it demands an intensive collaboration between the authors, editorial staff, and the scientific community. Finally, we did not evaluate the timeframe after retraction between the publication of the clinical trial and its retraction note.

## 5. Conclusions

It is necessary to constantly check information before taking it as trustworthy and truthful. We live in a time where information is only a few clicks away, which makes it amazingly easy to spread false and unverified information online. Scientific journals are among the safest spaces to seek and read quality and proven information. But, in a rush to publish COVID-19-related articles, in the heat of the pandemic, some low-quality articles with erroneous information were published, reaching thousands of people who trust journals to prove information before publication. Real quality science, evidence-based, can always be checked and proven. No “publish or perish” is worth the sacrifice and jeopardy of quality research. More and more articles are published every day. While only a few of them end up retracted, of those that are, their retraction notes need to obtain as many views as their original paper for people not to take erroneous information as truthful. In brief, as warned by Prof. Ambrosino to younger researchers (and the oldest ones): “*Reproducible and transparent procedures should be incorporated into research. Publications should provide sufficient information about materials, protocols, raw data, statistical analysis, and other indicators. Clinical decisions may depend on replicable or refutable results*” [25]. Then, we need to improve our efforts to perform a high-standard science because it is the best way to treat those under severe conditions, such as infection with SARS-CoV-2.

## Figures and Tables

**Figure 1 ijerph-20-01835-f001:**
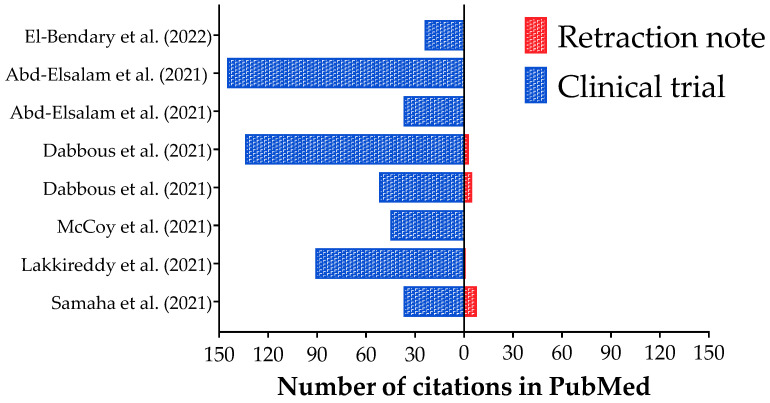
Number of citations in Pubmed database for the retracted clinical trials about Coronavirus Disease (COVID)-19 interventions and their retractions notes. References: [9,10,11,12,13,14,15,16].

**Figure 2 ijerph-20-01835-f002:**
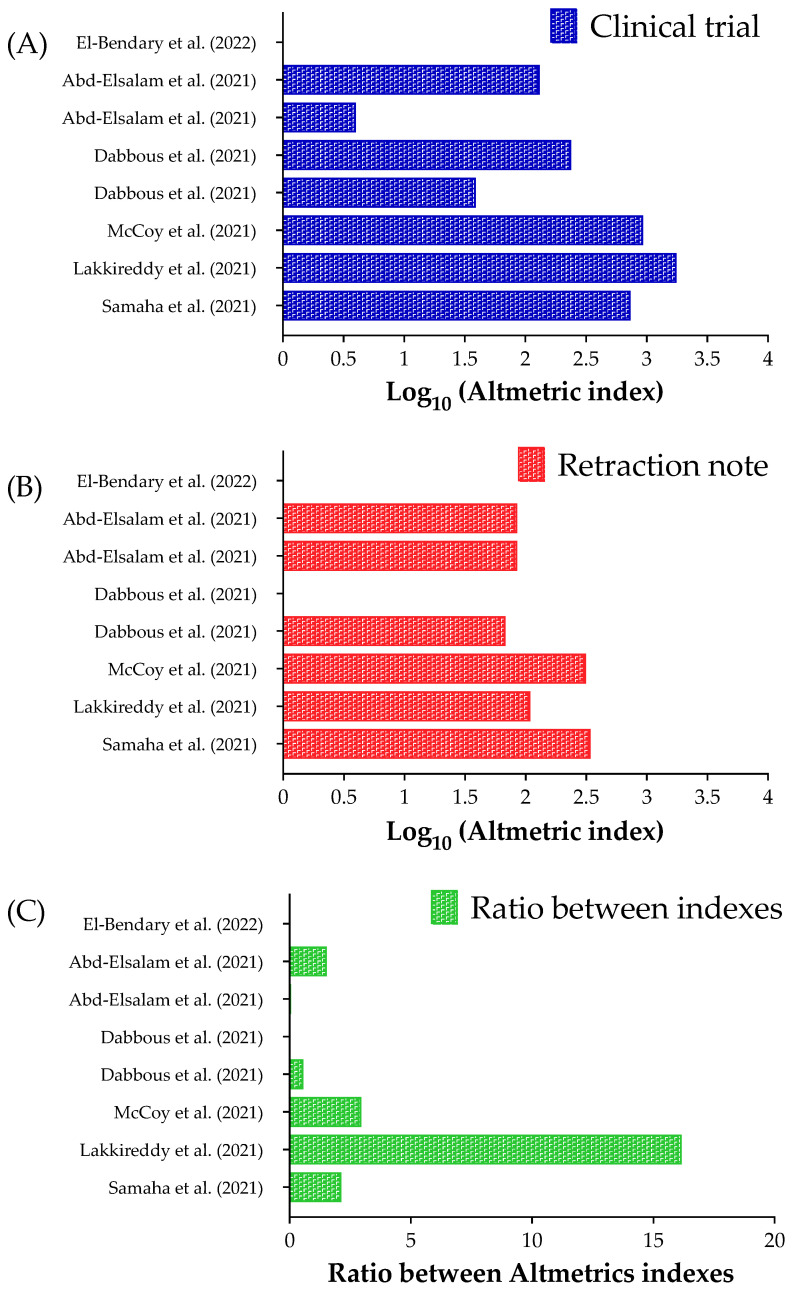
Altmetric index for the clinical trials and their retraction notes related to Coronavirus Disease (COVID-19) interventions. (**A**) Altmetric index for the clinical trials. (**B**) Altmetric index for the retraction notes. (**C**) The ratio between the Altmetric index for the clinical trials and their retraction notes. We presented the Altmetric index using a Log_10_ scale in (**A**,**B**). It was impossible to calculate the ratio between the Altmetrics indexes for two clinical trials and their retractions notes [13,16]. The Altmetric index was obtained using the “Altmetric it!” when logged in Pubmed for each clinical trial and its retraction note. References: [9,10,11,12,13,14,15,16].

**Figure 3 ijerph-20-01835-f003:**
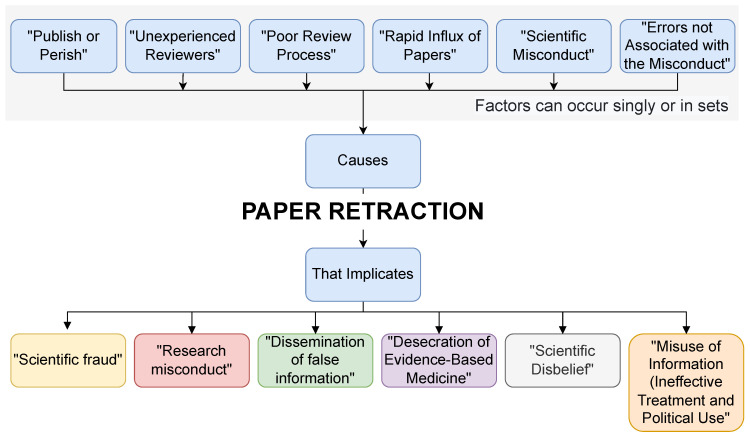
Paper retraction (causes and implications).

**Table 1 ijerph-20-01835-t001:** Number of clinical trials published in PubMed during the Coronavirus Disease (COVID)-19 pandemic and the proportion between COVID-19-related clinical trials and other clinical trials.

Study Type by Indexation in PubMed *	Clinical Trials (COVID-19-Related)—8445 Studies (A)	Clinical Trials (non-COVID-19-Related)—202,398 Studies (B) **	Proportion between the Percentages from (A) and (B) ***
Clinical trial phases			
I	142 (1.68%)	4774 (2.36%)	0.71
II	250 (2.96%)	7426 (3.67%)	0.81
III	156 (1.85%)	5445 (2.69%)	0.69
IV	17 (0.20%)	594 (0.29%)	0.69
Clinical trial protocol	549 (6.50%)	7942 (3.92%)	1.66
Randomized clinical trial	2086 (24.70%)	92,111 (45.51%)	0.54
Retracted clinical trials	8 (0.11%)	111 (0.05%)	1.73

* The sum of all types of clinical drugs did not represent the total amount of published clinical trials; ** We added the number of clinical trials for the entire 2019 year, and, in this case, the ratio between the retractions can be even higher; *** The proportion was calculated using the following formula: [Type of clinical trial—COVID-19-related/Clinical trials COVID-19 related—8445) by (Type of clinical trial—non-COVID-19-related/Clinical trials non-COVID-19 related—202,398)].

**Table 2 ijerph-20-01835-t002:** Title of the retracted studies, first author’s country, and journal where the study was published.

Study	First Author Country	Journal	Impact Factor
Effects of a single dose of Ivermectin on viral and clinical outcomes in asymptomatic SARS-CoV-2 infected subjects: A pilot clinical trial in Lebanon [9]	Lebanon	Viruses	5.818
Impact of daily high dose oral vitamin D therapy on the inflammatory markers in patients with COVID-19 disease [10]	India	Scientific Reports	4.996
Proxalutamide reduces the rate of hospitalization for COVID-19 male outpatients: a randomized double-blinded placebo-controlled trial [11] *	United States of America	Frontiers in Medicine	5.058
Safety and efficacy of Favipiravir versus Hydroxychloroquine in management of COVID-19: A randomized controlled trial [12]	Egypt	Scientific Reports	4.996
Efficacy of Favipiravir in COVID-19 treatment: a multi-center randomized study [13]	Egypt	Archives of Virology	2.685
Remdesivir efficacy in COVID-19 treatment: A randomized controlled trial [14]	Egypt	American Journal of Tropical Medicine and Hygiene	2.345
Hydroxychloroquine in the treatment of COVID-19: A multi-center randomized controlled study [15]	Egypt	American Journal of Tropical Medicine and Hygiene	2.345
Efficacy of combined Sofosbuvir and Daclatasvir in the treatment of COVID-19 patients with pneumonia: a multi-center Egyptian study [16]	Egypt	Expert Review of Anti-Infective Therapy	5.091

* The study was performed in Brazil.

**Table 3 ijerph-20-01835-t003:** Characterization of the clinical trials about Coronavirus Disease (COVID)-19 retracted after publication.

Study	Objective	Methods (Intervention, Study Design, and Outcomes)	Results and Conclusion	Retraction Notes	Altmetric *	
					Clinical Trial	Retraction Note
[9]	Determine the efficacy of Ivermectin, an Food and Drug Administration-approved drug, in producing clinical benefits and decreasing the viral load of SARS-CoV-2 among asymptomatic participants that tested positive for SARS-CoV-2 in Lebanon.	A randomized controlled trial was conducted on 100 asymptomatic Lebanese participants who tested positive for SARS-CoV-2. Fifty patients received standard preventive treatment, mainly supplements. The experimental group received a single dose (according to body weight) of Ivermectin and the same supplements the control group received.	72 h after the treatment regimen started, the increase in Ct-values was higher in the Ivermectin than in the control group. Moreover, more participants in the control group developed clinical symptoms. Three individuals (6%) from this study group required hospitalization—Ivermectin group (0%). In this context, Ivermectin appears to be efficacious in providing clinical benefits in a randomized treatment of asymptomatic SARS-CoV-2-positive participants, effectively resulting in fewer symptoms, lower viral load, and reduced hospital admissions.	After the publication, the authors contacted the editorial office regarding an error between files used for the statistical analysis. In adhering to the complaints procedure from the journal, an investigation confirmed the error reported by the authors. The Editor approved the retraction. Also, the authors agreed to this retraction.	736 (37 citations in PubMed)	344 (eight citations in PubMed)
[10]	Investigate the impact of Pulse D therapy in reducing the inflammatory markers of COVID-19.	Consented COVID-19 participants with hypovitaminosis D were evaluated for inflammatory markers [neutrophil/lymphocyte ratio, C-reactive protein, lactate dehydrogenase, Interleukin-6, and Ferritin] along with vitamin D on the first day and 9th/11th days as per their respective body mass index category. Subjects were randomized into vitamin D and non-vitamin D groups. The Vitamin D group received Pulse D therapy (targeted daily supplementation of 60,000 IUs of vitamin D for eight or ten days depending upon their body mass index) in addition to the standard treatment. Non-vitamin D group received standard therapy alone.	Eighty-seven out of one hundred and thirty subjects have completed the study (vitamin D: 44, and non-vitamin D: 43). Vitamin D level has increased from 16 ± 6 ng/mL to 89 ± 32 ng/mL after Pulse D therapy in vitamin D group and was associated with a significant reduction of the measured inflammatory markers; the reduction of these markers in non-vitamin D group was insignificant. The difference in the reduction of markers between the study groups was also significant.	After publication, concerns were raised about several aspects of the study: at baseline, there are significant differences in the parameters measured, indicating that randomization may not have been performed correctly. Post-publication peer review has confirmed that the alternative allocation method was inappropriate for randomized clinical trials. This means that the patients were not precisely randomized; therefore, the differences in outcome between the two arms cannot be attributed to the Pulse D therapy only.	1763 (91 citations in PubMed)	109 (one citation in Pubmed)
[11]	Determine if Proxalutamide, an androgen receptor antagonist, could be an effective treatment for men with COVID-19 in an outpatient setting.	A randomized, double-blinded, and a placebo-controlled clinical trial was conducted at two outpatient centers (Brasilia, Brazil). Male participants with confirmed COVID-19 but not requiring hospitalization (8-point ordinal scale < 3) were administered Proxalutamide 200 mg/day or placebo for up to seven days. The primary endpoint was the hospitalization rate at 30 days post-randomization. A total of 268 men were randomized in a 1:1 ratio: 134 received Proxalutamide, and 134 received a placebo. The participants were included in the intention-to-treat analysis.	The 30-day hospitalization rate was 2.2% in men taking Proxalutamide when compared to 26% in placebo. The 30-day hospitalization risk ratio was 0.09 (95%CI = 0.03–0.27). Patients in the Proxalutamide arm more frequently reported gastrointestinal adverse events; however, no patient discontinued treatment. In the placebo group, six patients were lost during follow-up, and two participants died from acute respiratory distress syndrome. The study demonstrated that the hospitalization rate in Proxalutamide-treated men was reduced by 91% when compared to usual care.	After publication, the journal received letters of complaint questioning the integrity of the article, following which an Expression of Concern was published. A thorough investigation was conducted following policies and Committee on Publication Ethics guidelines. The investigation found that the methods did not adequately support the claims made in the conclusions. Mainly, the allocation to treatment and control groups was not sufficiently random.	936 (45 citations in PubMed)	316 (No citation in PubMed)
[12]	Explore the safety and efficacy of Favipiravir in the treatment of COVID-19 mild and moderate cases.	The authors performed a randomized-controlled open-label interventional phase 3 clinical trial. One hundred patients were recruited from 18th April till 18th May. Fifty participants received Favipiravir 3200 mg on day 1, followed by 600 mg twice (day 2-day 10). Fifty participants received Hydroxychloroquine 800 mg on day 1, followed by 200 mg twice (days 2–10) and oral Oseltamivir 75 mg/12 h/day for ten days. Participants were enrolled in Ain Shams University Hospital and Assiut University Hospital. Both arms were comparable as regards demographic characteristics and comorbidities.	The average onset of SARS-CoV-2 PCR (polymerase chain reaction) negativity was 8.1 and 8.3 days in the Hydroxychloroquine-arm and Favipiravir-arm, respectively: 55.1% of those on the Hydroxychloroquine-arm turned PCR negative at/or before the 7th day from diagnosis when compared to 48% on the Favipiravir-arm. Four patients in the Favipiravir arm developed transient transaminitis. On the other hand, heartburn and nausea were reported in about 20 participants in Hydroxychloroquine-arm. Only one patient in Hydroxychloroquine-arm died after developing acute myocarditis, which resulted in acute heart failure. Favipiravir was considered a safe and effective alternative to Hydroxychloroquine in mild or moderate COVID-19-infected patients.	Concerns were brought to the Editors’ attention after publication, and the study’s raw data were requested. The authors provided several versions of their dataset. Post-publication peer review confirmed that none of these versions fully recapitulates the results presented in the cohort background comparisons, casting doubt on the reliability of the data. Additional concerns were raised about the randomization procedure, as the equal distribution of male and female patients is unlikely unless sex is a parameter considered during randomization. However, based on the clarification provided by the authors, sex was not considered during this process. The Editors, therefore, no longer have confidence in the results and conclusions presented.	39 (52 citations in PubMed)	68 (Five citations in PubMed)
[13]	Evaluate the efficacy of Favipiravir in the treatment of patients with COVID-19.	A multi-center, randomized, interventional phase 2/3 study that included participants with COVID-19 was performed. Ninety-eight participants were eligible to participate. After excluding participants who refused to participate, 96 participants were randomly assigned into two groups. The Chloroquine group included 48 participants who received Chloroquine 600 mg tablets twice daily added to the standard-of-care therapy for ten days. The Favipiravir group included 48 participants who received 1600 mg of Favipiravir twice a day on the first day and 600 mg twice daily from the second to the tenth day, added to the standard-of-care therapy for ten days. Four participants in this group quit after the beginning of the study, and the final number in this group was 44. The four patients who left the study preferred to complete their treatment and be transferred to military hospitals, after which the authors lost contact with them.	Although not statistically significant, the Favipiravir group had a lower mean duration of hospital stay than the Chloroquine group (13.29 ± 5.86 vs. 15.89 ± 4.75 days). None of the participants in the Favipiravir group needed mechanical ventilation or had an oxygen saturation < 90%, but these differences were insignificant when compared to the Chloroquine group. Four patients in the Chloroquine group required mechanical ventilation and received Methylprednisolone after their condition worsened. Two patients (4.2%) in the Chloroquine group and one (2.3%) in the Favipiravir group died; however, no significant differences were observed between the groups regarding side effects. The patient’s age and C-reactive protein level were the only factors significantly associated with mortality, and Favipiravir treatment was not significantly related to COVID-19 mortality. The authors concluded that Favipiravir is a promising drug for the treatment of COVID-19 that might decrease the hospital stay and the need for mechanical ventilation.	After publication, concerns were raised about reporting this clinical trial, and the authors were asked to provide their raw data files. The raw data was examined. First, the reported baseline variables showed that the distribution of one variable was highly statistically different in the two study groups. Second, two variables showed different rounding to significant figures in the two groups. Third, for two variables, there was a different distribution of the variables when moving through the groups. It is unclear how these variations could occur in a correctly performed trial, so severe doubts about the randomization process and data validity arose. These doubts were reinforced by the equal sex distribution even though sex was not stated as an inclusion parameter.	239 (134 citations in PubMed)	0 (Three citations in PubMed)
[14]	To assess the efficacy of Remdesivir in hospitalized Egyptian patients with COVID-19.	Patients were randomly assigned at a 1:1 ratio to receive either Remdesivir in addition to standard care or standard care alone. Two hundred patients (100 in each group) completed the study and were included in the final analysis.	The Remdesivir group showed a significantly lower median duration of hospital stay than the control group (10 vs. 16 days). Eleven patients in the Remdesivir group needed mechanical ventilation when compared with eight patients in the control group. The mortality rate was comparable between the two groups; however, it was significantly associated with older age, elevated C-reactive protein levels, elevated D-dimer, and the need for mechanical ventilation. Remdesivir positively influenced the length of hospital stay, but it had no mortality benefit.	Data overlap and copying were described. Second, the authors have not provided a reasonable explanation for this significant problem. Third, adequate data error-checking or validation has not been provided to ensure that the results presented in the paper accurately represent the sourced data. The journal lost confidence in the totality of the data. Because of this and insufficient answers to repeated inquiries, the journal retracted the article.	Four (37 citations in PubMed)	85 (No citation in PubMed)
[15]	The authors aimed to evaluate the safety and efficacy of Hydroxychloroquine added to standard care in patients with COVID-19.	A multi-center, randomized controlled trial was conducted at three major university hospitals in Egypt. One hundred ninety-four patients with a confirmed diagnosis of COVID-19 were included. They were equally randomized into two arms: 97 patients administrated Hydroxychloroquine plus standard care, and 97 patients administered only standard care as a control arm. The primary endpoints were recovery within 28 days, need for mechanical ventilation, or death. The two groups were matched for age and sex, and there was no significant difference between them regarding baseline characteristics or laboratory parameters.	Four patients (4.1%) in the Hydroxychloroquine group and five (5.2%) in the control group needed mechanical ventilation. The overall mortality did not differ between the two groups, as six patients (6.2%) died in the Hydroxychloroquine group, and five (5.2%) died in the control group. Univariate logistic regression analysis showed that Hydroxychloroquine treatment was not significantly associated with decreased mortality in patients with COVID-19. So, adding Hydroxychloroquine to standard care did not add significant benefit, did not decrease the need for ventilation, and did not reduce mortality rates.	Data overlap and copying were described. Second, the authors have not provided a reasonable explanation for this significant problem. Third, adequate data error-checking or validation has not been provided to ensure that the results presented in the paper accurately represent the sourced data. The journal lost confidence in the totality of the data. Because of this and insufficient answers to repeated inquiries, the journal retracted the article.	131 (145 citations in PubMed)	85 (No citation in PubMed)
[16]	The study evaluated the efficacy of generic Sofosbuvir/Daclatasvir in treating patients with COVID-19 who presented with pneumonia.	This multi-center prospective study involved 174 patients with COVID-19. Patients were randomized into two groups. Group A (96 patients) received Sofosbuvir (400 mg)/Daclatasvir (60 mg) for 14 days in combination with conventional therapy. Group B (78 patients) received conventional therapy alone. Clinical, laboratory, and radiological data were collected at baseline and after 7, 14, and 28 days of therapy. The primary endpoint was the rate of clinical/virological cure.	A lower mortality rate was observed in Group A (14 vs. 21%). After one month of therapy, no differences were found in intensive care unit admission rates, oxygen therapy, or ventilation. Additionally, a statistically significant shorter duration of hospital stay (9 vs. 12%) and a faster achievement of PCR negativity at day 14 (84 vs. 47%) were noticed in Group A.	Since publication, concerns were raised about the integrity of the data and reported results. When approached for an explanation, the authors could not address the concerns raised and could not provide sufficient supporting information. As verifying the validity of published work is core to the integrity of the scholarly record, the article was retracted. The corresponding author listed in this publication was informed. The authors disagree with the retraction.	One (24 citations in PubMed)	0 (No citation PubMed)

IUs, international units; 95%CI, 95% confidence interval; %, percentage; Ct, number of cycles necessary to spot the virus; SARS-CoV-2, Severe Acute Respiratory Syndrome Coronavirus 2; mL, milliliters; mg, milligrams; ng, nanograms; *N*, number of individuals (participants). * We presented the crude data in table and log10 in Figure 2a,b.

## Data Availability

Not applicable.

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
