# Peer review of "Retraction of Clinical Trials about the SARS-CoV-2 Infection: An Unaddressed Problem and Its Possible Impact on Coronavirus Disease (COVID)-19 Treatment"

_ijerph, 2023, doi:10.3390/ijerph20031835_

Round 1
Reviewer 1 Report
Thank you for the opportunity to review this manuscript on the retraction of papers during the COVID-19 pandemic and the need to prioritize efforts in assuring quality evidence-based publications.
I have the following comments for the authors to consider:
-The study’s title does not clearly suggest that the retraction of the clinical trials was correlated with the COVID-19 pandemic.
-It is also not clear in the abstract, introduction, and methods why the authors chose the COVID-19 period (i.e., why not compare retractions before and during?), the relevance of the study on COVID-19 treatment, and why the retraction of studies puts science in jeopardy.
-Is the focus on clinical trials for COVID-19 interventions or all interventions during the pandemic?
- In line 89, the authors mentioned that they present the characterization of clinical trials about COVID-19. There is an included study using ivermectin to control viral load and another on a drug to manage emotional disorders in patients with DM during the pandemic. Please clearly elaborate on the study’s objectives and the criteria used to assess the eligibility of the included studies.
-Please justify why PubMed was the only database searched.
-The authors mention in their study that retraction is high during the pandemic and cited references. I suggest the authors report the number of retractions reported in these studies and discuss the similarities and differences in context.
-Although they mentioned that the number of retractions is high (first sentence in discussion), they wrote that the nine studies retracted are low in the limitations. Please clarify.
Author Response
Review Report Form
Reviewer 1.
Comments and Suggestions for Authors
Thank you for the opportunity to review this manuscript on the retraction of papers during the COVID-19 pandemic and the need to prioritize efforts in assuring quality evidence-based publications.
I have the following comments for the authors to consider:
-The study’s title does not clearly suggest that the retraction of the clinical trials was correlated with the COVID-19 pandemic.
Reply: The authors thank the reviewer. In this context, we corrected the title as following: “Retraction of Clinical Trials about the SARS-CoV-2 Infection: An Unaddressed Problem and its Possible Impact on Corona-virus Disease (COVID)-19 Treatment”.
-It is also not clear in the abstract, introduction, and methods why the authors chose the COVID-19 period (i.e., why not compare retractions before and during?), the relevance of the study on COVID-19 treatment, and why the retraction of studies puts science in jeopardy.
Reply: Dear reviewer, we performed several corrections in the study, mainly in the results section. Now, we believe that the study is better presented. In addition, we can include other corrections if necessary.
-Is the focus on clinical trials for COVID-19 interventions or all interventions during the pandemic?
Reply: The authors thank the reviewer. We added several corrections in the text. In brief, we analyzed the retractions on clinical trials for COVID-19 interventions. In addition, during the revision, we include only the number of all interventions during the pandemic (excluding the COVID-19-related) and we described the number of retractions for these papers and we performed a comparison between both groups of papers. Please, we presented the complete corrections, mainly, in methods and results sections.
- In line 89, the authors mentioned that they present the characterization of clinical trials about COVID-19. There is an included study using ivermectin to control viral load and another on a drug to manage emotional disorders in patients with DM during the pandemic. Please clearly elaborate on the study’s objectives and the criteria used to assess the eligibility of the included studies.
Reply: Dear reviewer, we corrected the objectives and we excluded the description of the manuscript that discuss the management of emotional disorders in patients with diabetes mellitus during the COVID-19 pandemic.
-Please justify why PubMed was the only database searched.
Reply: The authors thank the reviewer for the important contribution. In this context, we included the following statement in the methods section: “PubMed database was selected to standardize the study because PubMed database uses an algorithm that searches the title, abstract, and headings of articles in the National Library of Medicine database – specific to medicine and health [26].”
-The authors mention in their study that retraction is high during the pandemic and cited references. I suggest the authors report the number of retractions reported in these studies and discuss the similarities and differences in context.
Reply: We included the number of retractions for the clinical trials performed during the same period and we compared both groups. Please, the complete corrections are presented in the methods and results sections.
-Although they mentioned that the number of retractions is high (first sentence in discussion), they wrote that the nine studies retracted are low in the limitations. Please clarify.
Reply: Dear reviewer we corrected the text and the limitations section: “The main limitations of our study included the following topics: The authors only ana-lyzed eight retracted articles, which is a low number when compared to the immense number of published articles during the COVID-19 pandemic. But, despite the low number of retracted articles analyzed, this paper calls for attention to a severe problem. Also, proportionally, there is a double of papers related to COVID-19 interventions than clinical trials non-COVID-19 related during the same period. Moreover, very little in-formation was available regarding the retraction note and its justifications. In addition, it is difficult to assess the quality of clinical trials, including randomized controlled trials ones, after publication, and it demands an intensive collaboration between the authors, editorial staff, and the scientific community.”

Reviewer 2 Report
This article exacerbates the problem of retracted articles receiving citations by citing the retracted articles.
Line 51-52 – It is unclear if the assertation that “impact factor of the journals can reflect the quality of the journal and, by extension, the papers it publishes,” is supported by data. Please clarify.
In figure 1, the final line seems to have a lot of euphemisms or buzzwords (e.g. paperdemic, fake news etc.) that could be replaced either “scientific fraud” or “research misconduct”. Characterization is poorly supported by this paper where about half of the articles were retracted for suspected research misconduct while others were retracted for reasons not addressed in the figure. The figure also creates a sense that 1) this was due to pressure to publish, which has not been demonstrated empirically for any of these articles. 2) That retractions are due to insufficient peer review, which implies that skilled reviewers can detect randomization errors, research misconduct, and similar small errors resulting in work that is not reproducible. It is unclear if skilled reviewers would request raw data or be able to detect some of the issues identified as the cause of retractions upon review. 3) That the rapid influx of papers causes retractions. The authors have not demonstrated that the retraction rate increased compared to times when less papers were submitted. Overall, this figure could be omitted or revised to better support the data presented.
Line 217-219 – This highlights a major weakness of the research. The retraction rate (e.g. number of retractions per published paper) for clinical trials is not investigated in the manuscript. Therefore, conclusions about whether these COVID-19 related clinical trial retractions compared to the normal baseline retraction level of all clinical research cannot be determined.
Overall, the premises of the article, that published articles gain more attention than retractions, is somewhat misguided because it seems to be based on citation and sharing. Once the article is retracted, researchers generally stop citing the article completely rather than citing the retraction. A better measure would be to look at the number of citations of the article after the retraction became available (probably including the timeframe after retraction) to show whether or not the articles continue to be cited.
The introduction has a lot of conjecture and could be improved with a more thorough review of the science of retracted articles and their citations. See:
Fang, F. C., & Casadevall, A. (2011). Retracted science and the retraction index. Infection and immunity, 79(10), 3855-3859.
Bolland, M. J., Grey, A., & Avenell, A. (2022). Citation of retracted publications: A challenging problem. Accountability in Research, 29(1), 18-25.
Bar-Ilan, J., & Halevi, G. (2018). Temporal characteristics of retracted articles. Scientometrics, 116(3), 1771-1783.
Marcus, A., Abritis, A. J., & Oransky, I. (2022). How to stop the unknowing citation of retracted papers. Anesthesiology, 137(3), 280-282.
The higher Altmeter index seems to be an artifact of citations of the retracted article. It is unclear how the altimeter score is calculated.
Minor comments
The columns on Table 2 cause many lines of text to be split without hyphenation.
The Altmeter and PubMed citations from table 2 would be easier to interpret in a graphical format.
Line 46 – “de” should be “the”
The paper has numerous examples of awkward phrasing and grammar. English language editing would improve the readability.
Author Response
Reviewer 2.
This article exacerbates the problem of retracted articles receiving citations by citing the retracted articles.
Line 51-52 – It is unclear if the assertation that “impact factor of the journals can reflect the quality of the journal and, by extension, the papers it publishes,” is supported by data. Please clarify.
Reply: The authors thank the reviewer. We excluded the following statement from the text: “For that matter, the impact factor of the journals can reflect the quality of the journal and, by extension, the papers it publishes.”
In figure 1, the final line seems to have a lot of euphemisms or buzzwords (e.g. paperdemic, fake news etc.) that could be replaced either “scientific fraud” or “research misconduct”.
Reply: We changed the words as recommended.
Characterization is poorly supported by this paper where about half of the articles were retracted for suspected research misconduct while others were retracted for reasons not addressed in the figure. The figure also creates a sense that 1) this was due to pressure to publish, which has not been demonstrated empirically for any of these articles. 2) That retractions are due to insufficient peer review, which implies that skilled reviewers can detect randomization errors, research misconduct, and similar small errors resulting in work that is not reproducible. It is unclear if skilled reviewers would request raw data or be able to detect some of the issues identified as the cause of retractions upon review. 3) That the rapid influx of papers causes retractions. The authors have not demonstrated that the retraction rate increased compared to times when less papers were submitted. Overall, this figure could be omitted or revised to better support the data presented.
Reply: We revised the figure 1 and we added the following excerpt in the text:
“One question remains: why were the retraction rates on COVID-19 papers higher during the pandemic? After consideration, the authors ended up with three possibilities for a reason for this phenomenon:
1) this was due to pressure to publish, which, despite not being demonstrated empirically for any of these articles, is a valid possibility and further studies are to elucidate this possibility.
2) That retractions are due to insufficient peer review, which implies that skilled reviewers can detect randomization errors, research misconduct, and similar minor errors resulting in work that is not reproducible. It is unclear whether skilled and experienced reviewers would request raw data from the studies or be able to detect some of the issues identified as the cause of retractions upon review.
3) That the rapid influx of papers causes retractions. In this sense, due to the urgency for new information and possible treatment for a new deadly disease, all researchers turned to COVID-19. This converged in a high number of articles being submitted simultaneously.”
Line 217-219 – This highlights a major weakness of the research. The retraction rate (e.g. number of retractions per published paper) for clinical trials is not investigated in the manuscript. Therefore, conclusions about whether these COVID-19 related clinical trial retractions compared to the normal baseline retraction level of all clinical research cannot be determined.
Reply: Dear reviewer, we added the information in the study. Thank you to collaborate improving our study.
Overall, the premises of the article, that published articles gain more attention than retractions, is somewhat misguided because it seems to be based on citation and sharing. Once the article is retracted, researchers generally stop citing the article completely rather than citing the retraction. A better measure would be to look at the number of citations of the article after the retraction became available (probably including the timeframe after retraction) to show whether or not the articles continue to be cited.
Reply: Dear reviewer, we corrected the text based on this comment. Please, to see the new version of the manuscript.
The introduction has a lot of conjecture and could be improved with a more thorough review of the science of retracted articles and their citations. See:
Fang, F. C., & Casadevall, A. (2011). Retracted science and the retraction index. Infection and immunity, 79(10), 3855-3859.
Bolland, M. J., Grey, A., & Avenell, A. (2022). Citation of retracted publications: A challenging problem. Accountability in Research, 29(1), 18-25.
Bar-Ilan, J., & Halevi, G. (2018). Temporal characteristics of retracted articles. Scientometrics, 116(3), 1771-1783.
Marcus, A., Abritis, A. J., & Oransky, I. (2022). How to stop the unknowing citation of retracted papers. Anesthesiology, 137(3), 280-282.
Reply: We added the citations and we included several corrections in the text.
The higher Altmeter index seems to be an artifact of citations of the retracted article. It is unclear how the altimeter score is calculated.
Reply: Dear reviewer, we added the information:
“In this context, considering only the retracted papers for COVID-19-related inter-ventions, we collected the information for i) the first author’s country; ii) the Journal name where the study was published; iii) the impact factor of the journal; iv) the main objective of the study; v) methods including population, intervention, study design, and outcomes; and vi) results and conclusions. Also, we collected complete information from the re-traction notes published by the journals. We also included the Altmetric index for the clinical trials and the retraction notes to compare the accessibility to both studies’ indexes (https://www.altmetric.com/). In addition, the number of citations achieved from the PubMed database was described.
The Altmetric is an index that collects and collates all of the disparate information about research to provide the Scientific community with a visually engaging and in-formative view of the online activity surrounding the scholarly content. In brief, as de-scribed by the developers, “Altmetrics are metrics and qualitative data that are complementary to traditional, citation-based metrics. They can include (but are not limited to) peer reviews on Faculty of 1000, citations on Wikipedia and in public policy documents, discussions on research blogs, mainstream media coverage, bookmarks on reference managers like Mendeley, and mentions on social networks such as Twitter. Sourced from the Web, Altmetrics can tell you a lot about how often journal articles and other scholarly outputs like datasets are discussed and used around the world. For that reason, Altmetrics have been incorporated into researchers’ websites, institutional repositories, journal websites, and more.”
Minor comments
The columns on Table 2 cause many lines of text to be split without hyphenation.
Reply: We corrected the Table, and we did not notice more words without hyphenation.
The Altmeter and PubMed citations from table 2 would be easier to interpret in a graphical format.
Reply: The authors the reviewer and we added two new figures.
Line 46 – “de” should be “the”
Reply: We corrected the text.
The paper has numerous examples of awkward phrasing and grammar. English language editing would improve the readability.
Reply: We corrected the text.
